Contribution of insect pollinators to crop yield and quality varies with agricultural intensification

Bartomeus Ignasi 1 nacho.bartomeus@gmail.com
Potts Simon G. 2
Steffan-Dewenter Ingolf 3
Vaissière Bernard E. 4
Woyciechowski Michal 5
Krewenka Kristin M. 6
Tscheulin Thomas 2 7
Roberts Stuart P.M. 2
Szentgyörgyi Hajnalka 5
Westphal Catrin 6
Bommarco Riccardo 1
1 Department of Ecology, Swedish University of Agricultural Sciences , Uppsala , Sweden
2 School of Agriculture, Policy and Development, University of Reading , Reading , UK
3 Department of Animal Ecology and Tropical Biology, Biocentre, University of Würzburg , Würzburg , Germany
4 INRA, UR406 Abeilles & Environnement , Avignon , France
5 Institute of Environmental Sciences, Jagiellonian University , Gronostajowa, Krakow , Poland
6 Department of Crop Science, Agroecology, Georg-August-University , Göttingen , Germany
7 Department of Geography, University of the Aegean , Mytilene , Greece
Traveset Anna
Electronic publication date: 2014 Mar 27
Publication date: 2014
Volume: 2
Electronic Location ID: e328
Received 2014 Jan 2; Accepted 2014 Mar 12
Copyright: © 2014 Bartomeus et al.
Copyright year: 2014
Copyright holder: Bartomeus et al.
License: This is an open access article distributed under the terms of the Creative Commons Attribution License, which permits unrestricted use, distribution, and reproduction in any medium, provided the original author and source are credited.
License URL: https://creativecommons.org/licenses/by/3.0/

Keywords: Biodiversity, Pollination, Honeybees, Wild bees, Agroecosystems, Ecosystem services

Funding: EU 6th Framework Programme 2003-506675 (ALARM) EU 7th Framework Programme 244090 (STEP) This work was funded by the European Union to the projects “ALARM—Assessing large-scale environmental risks for biodiversity with tested methods” (2003-506675), and “STEP—Status and trends of European pollinators” (244090) in the 6th and 7th Framework Programme, respectively, and by the Swedish research council FORMAS. The funders had no role in study design, data collection and analysis, decision to publish, or preparation of the manuscript.

==============================
Background. Up to 75% of crop species benefit at least to some degree from animal pollination for fruit or seed set and yield. However, basic information on the level of pollinator dependence and pollinator contribution to yield is lacking for many crops. Even less is known about how insect pollination affects crop quality. Given that habitat loss and agricultural intensification are known to decrease pollinator richness and abundance, there is a need to assess the consequences for different components of crop production.

Methods. We used pollination exclusion on flowers or inflorescences on a whole plant basis to assess the contribution of insect pollination to crop yield and quality in four flowering crops (spring oilseed rape, field bean, strawberry, and buckwheat) located in four regions of Europe. For each crop, we recorded abundance and species richness of flower visiting insects in ten fields located along a gradient from simple to heterogeneous landscapes.

Results. Insect pollination enhanced average crop yield between 18 and 71% depending on the crop. Yield quality was also enhanced in most crops. For instance, oilseed rape had higher oil and lower chlorophyll contents when adequately pollinated, the proportion of empty seeds decreased in buckwheat, and strawberries’ commercial grade improved; however, we did not find higher nitrogen content in open pollinated field beans. Complex landscapes had a higher overall species richness of wild pollinators across crops, but visitation rates were only higher in complex landscapes for some crops. On the contrary, the overall yield was consistently enhanced by higher visitation rates, but not by higher pollinator richness.

Discussion. For the four crops in this study, there is clear benefit delivered by pollinators on yield quantity and/or quality, but it is not maximized under current agricultural intensification. Honeybees, the most abundant pollinator, might partially compensate the loss of wild pollinators in some areas, but our results suggest the need of landscape-scale actions to enhance wild pollinator populations.

Introduction

There is growing evidence that ecosystem services, such as biological pest control and crop pollination, benefit food production (Bommarco, Kleijn & Potts, 2013). Indeed, 75% of the crop species used for food depend on insect pollination to some degree (Klein et al., 2007). More than a decade of active pollination research has led to a greatly improved general understanding on animal pollination benefits to crop yields worldwide (e.g., Klein et al., 2007; Garibaldi et al., 2011; Garibaldi et al., 2013). However, major knowledge gaps remain.

First, we have surprisingly little information on the actual degree of pollinator dependence for some major crops. While some crops depend entirely on insect pollinator visits to set fruit, many others are only partly dependent on animal pollination and can produce more than 90% of the maximum seed or fruit yield without pollinators (Klein et al., 2007). The role of pollinators for crop production has mainly been examined in observational studies, relying primarily on natural variation in visitation rates among observed sites. Experiments directly manipulating insect flower visitation (e.g., excluded pollinators vs. open access of pollinators) are less common for most crops (but see Klein, Steffan-Dewenter & Tscharntke, 2003; Höhn et al., 2008). Assessing pollination dependence with proper controls is needed to correctly estimate the contribution that insect pollinators can provide to crop yields.

Second, most available studies quantify the number of fruits per plant. Fruit number can be a good proxy for yield (Garibaldi et al., 2013), which is the amount of produce harvested per unit area. However, the correlation between the number of fruit produced and yield may be low in some crops. For example, interspecific plant competition can lead to high variability in plant size and thereby fruit production among plants. This is especially critical for crops with indeterminate flowering and a high compensation capacity such as soybean (Glycine max) and oilseed rape (Brassica napus). For these, fruit set measured on a limited number of isolated plants is unlikely to be representative of the real production in a crop stand (Stivers & Swearingin, 1980; Angadi et al., 2003). Moreover, plants can allocate resources for producing fruits of variable size based on the number of fruits per plant and the level of pollination received (e.g., Gonzalez, Coque & Herrero, 1998 in kiwifruit Actinidia deliciosa), such that similar levels of fruit set can differ in total crop yield because of difference in fruit size (Bos et al., 2007). Again, the use of proper control plants from which pollinators are excluded is a way to better estimate the actual contribution of pollinators to yield in such crops.

Quality is also important in crop production, especially from an economic standpoint. Fruit quality can be negatively correlated with quantity when the fruit load on a tree or a vine is too high (e.g., Ferguson & Watkins, 1992 in apple Malus x domestica), but it is not so otherwise, especially in crops with indeterminate flowering such as oilseed rape (Bommarco, Marini & Vaissière, 2012). Indeed, adequate pollination often leads to produce with enhanced quality in entomophilous crops such as orchard fruit production (e.g., in apple—Garratt et al., 2013), as well as in field crops (oilseed rape—Bommarco, Marini & Vaissière, 2012) and small fruits and vegetables (e.g., strawberry Fragaria x ananassa—Andersson, Rundlöf & Smith, 2012; Chagnon, Gingras & Deoliveira, 1993; Roselino et al., 2009; tomato Solanum lycopersicum—Hogendoorn, Bartholomaeus & Keller, 2010; bell peppers Capsicum annuum—Roldan Serrano & Guerra-Sanz, 2006; highbush blueberry Vaccinium corymbosum—Isaacs & Kirk, 2010).

Given the drastic shifts in community composition of insects that visit flowering crops (Winfree, Bartomeus & Cariveau, 2011; Bommarco et al., 2011; Bartomeus et al., 2013a), and declines in numbers of pollinator species observed in some regions (Potts et al., 2010; Carvalheiro et al., 2013), it is increasingly important to gather information on the extent to which different crops depend on insect pollination for yield, and if current pollinator communities fulfill the demand for pollination services such that both crop quality and yields are maximized (Breeze et al., 2011). Relationships between land use intensity, pollinator visitation, and fruit set have been well studied. While pollinator species richness consistently and drastically decays as agricultural landscapes are deprived of natural habitat and are more intensively cultivated (Kennedy et al., 2013), this relationship is much weaker for fruit set (Garibaldi et al., 2011; Chacoff, Aizen & Aschero, 2008; Ricketts et al., 2008). One explanation for this difference is that the remaining pollinators provide sufficient visitation even in homogenous, intensively cultivated landscapes, especially if the crop has a large degree of autonomous self-pollination. Moreover, intensive landscapes are characterized by harboring just a few generalist pollinator species (Bartomeus & Winfree, 2013), but these might be in sufficient numbers to deliver enough crop pollination services. In fact, not all pollinator species respond equally to land use change (Williams et al., 2010; Winfree, Bartomeus & Cariveau, 2011), and some even increase in abundance with agricultural intensification (Westphal, Steffan-Dewenter & Tscharntke, 2003; Carré et al., 2009). This diversity of pollinator responses can, in some cropping systems, buffer a loss of pollination functioning (Cariveau et al., 2013); especially so if the pollinators who are the main ecosystem service providers are adapted to the ephemeral and patchy resource distribution that is typical for agricultural landscapes. Moreover, although wild insects increase fruit set independently of honeybee visits (Garibaldi et al., 2013), honeybees are less dependent on landscape characteristics because they are mainly managed, particularly in North America and Europe, and can be moved around in the landscape. Hence, honeybees can also help mitigate against wild pollinator loss in more intensively used landscapes where pollination services are degraded. In any case, the composition of the landscape in which the flowering crop field is embedded emerges as an important driver for pollinator community composition, and the landscape context needs to be considered when linking land use to pollination provisioning and benefits in field crops.

Here we used pollinator exclusion on the flowers or inflorescence on a whole plant basis in a set of crops under standard field conditions, to quantify pollinator dependency for four economically important annual crops in Europe. We assessed pollinator contribution to both yield quantity and quality. By replicating this experiment along a landscape gradient for each crop, we were able to test the hypothesis that pollinator visitation rate decrease with agricultural intensification and its consequences for crop pollination services and production.

Material and Methods

Study sites

The fieldwork was conducted in four European countries during May–August 2005 (Table 1). Spring oilseed rape (Brassica napus) was assessed in the region around the city of Uppsala, Sweden (see Bommarco, Marini & Vaissière, 2012, for details); field bean (Vicia faba) in around Reading, UK, strawberry (Fragaria x ananassa) around Göttingen, Germany; and buckwheat (Fagopyrum esculentum) near Krakow, Poland. For each crop, we selected ten fields that were separated by a minimum distance of 3 km, corresponding to the maximum foraging range of most bees (Greenleaf et al., 2007). Within each field, we established a 50∗25 m study area (5∗150 m for buckwheat as the fields were long and narrow) with a homogeneous and continuous crop cover. For fields up to two ha in size, this study site was located in the middle of the field. For larger fields, it was located between the geometric center of the field and one of its margins (Vaissière, Freitas & Gemill-Herren, 2011).

Table 1 Characteristics of the four study systems.

For each crop, the variety used, the distance between fields, field size and mean pollinator richness is shown.

	Variety	Distance between
sites (range in km)	Field sizes
(range in ha)	Mean species
richness of
pollinators	Percentage agriculture
in 500 m buffer
(range)	Percentage agriculture
in 1000 m buffer
(range)	
Oil seed rape	Stratos	3–7	1.0–40.4	11.3	14–50	4–44	
Field bean	Clipper	3–18	5.0–47.0	3.1	34–99	35–99	
Strawberry	Honeoye,
Korona,
Darselect,
Symphonie	3–26	0.3–1.3	12.9	51–99	48–96	
Buckwheat	Kora	4–7	0.3–4.0	11.4	29–82	41–73	

Insect sampling

In each field, we assessed the abundance and species richness of the major groups of flower-visiting insects, including bees (Hymenoptera: Apoidea: Apiformes), hoverflies (Diptera: Syrphidae), and butterflies (Lepidoptera). We used standardized transect walks with an aerial net (Westphal et al., 2008). In each study site, a 150 m transect line was established in the field near the experimental plots. An observer walked this line for 30 min identifying visiting insects at species level and catching unidentified species within a corridor 4 m wide. We performed the transect walks between 0900 and 1700 h only on days with temperatures at or above 15 °C, with no precipitation, dry vegetation, and low windspeed (<40 km h−1; Westphal et al., 2008). Specimens were pinned, labeled, and subsequently identified to species level. We returned four times to each study site during the main flowering period of each study crop.

Experimental design and yield analysis

In each of the ten fields, we established a block experiment with four blocks (Fig. 1). Each block had two treatments with one plot per treatment and five to ten tagged contiguous plants monitored per plot. The first treatment (Open) was open pollinated with all the flowers of each plant accessible to autonomous self-, wind- and insect-pollination. In the second treatment (Net), all flowers were enclosed in nylon tulle bags with 1∗1 mm openings (Diatex F510; http://www.diatex.fr/-Agriculture-.html) of an appropriate size to cover an inflorescence (buckwheat, field bean & oilseed rape), or an individual flower (strawberry). Thus, in the Net-treatment all flowers were exposed to wind- and self-pollination, but not to insect pollination. Because such nets do not hinder the airborne pollen flow (Sacchi & Price, 1988; Wragg & Johnson, 2011), the difference between these treatments represents the contribution from insect pollination. Bag manipulations were done carefully and in most cases before or after anthesis to avoid increased levels of self-pollination. We put the nets over the flower buds before the onset of flowering. Leaves and plant parts with no flowers were left as much as possible outside the net bag to minimize any effects of the bag on the photosynthesis (Howpage, Spooner Hart & Vithanage, 2001). As soon as flowers had wilted, we removed the nets, and the tagged plants were left to ripen in the field until harvest.

Figure 1 Experimental design.

Schema of the experimental design replicated in each of the 10 fields per crop showing the four blocks with two treatments each block and the 150 m transect surrounding the blocks.

For buckwheat, field bean, and oilseed rape, we cut all experimental plants from each plot and stored them individually in a linen bag just before commercial harvest. For strawberry, we followed the commercial harvest procedure and harvested the ripe strawberries twice a week. In each plot, we recorded fruit set as the number of fruits per plant (field bean, oilseed rape, and strawberry) or seed set as the number of seeds per plant (buckwheat). Yield was measured as the total weight of seeds per plant (buckwheat, field bean, and oilseed rape) or total fresh weight (strawberries) measured using a precision scale. As plants grew in field conditions with typical densities, the mean production per plant is a good proxy of tones per hectare obtained by the farmer. For each crop, we also measured the specific attributes of quality that affect its marketing value. For oilseed rape, we analyzed the oil content and chlorophyll contents of the seeds (performed by Svalof Weibull Lab AB, Svalov, Sweden). High chlorophyll contents decrease the durability and alter the color of the extracted oil. For field beans, we measured the nitrogen content of the seed as a proxy of their protein content. The nitrogen content was measured using oxidative combustion in an automated Dumas type combustion analyzer. For strawberry, we classified commercial quality as grade 1 (fully developed fruits of good quality), grade 2 (marketable fruits with some changes in colour and shape) and grade 3 (non-marketable fruits) according to guidelines of the German board of trade. For buckwheat, we measured the proportion of filled seeds since high proportion of empty seeds leads to a penalty in the market price. For buckwheat, six fields where destroyed due to a hailstorm, and hence we do not have yield measures for those.

Landscape context

The ten fields for each crop were located along a gradient of surrounding landscape complexity. The gradient ranged from intensive agricultural landscapes dominated by large arable fields with few boundary features, to complex landscapes with smaller average arable field sizes and more than 40% coverage of semi-natural habitats, such as pastures and forest patches over 0.5, 1, 2, and 3 km radius around each study field. When selecting the field sites, the proportion of arable land in the surrounding landscape was measured around each experimental field and used as a proxy for landscape complexity (Steffan-Dewenter et al., 2002; Fahrig, 2013). The proportion of arable land in the landscape surrounding each of the ten experimental fields varied depending on the region, with some regions presenting more intense landscapes (e.g., range of 48–97% of agricultural land for oilseed rape fields at 1000 m radius), and other regions presenting more complex landscapes (range of 4–45% of agricultural land for field bean at 1000 m radius; Table 1).

For oilseed rape, we used the Swedish digitized land cover terrain map database to characterize the landscape surrounding each field (Lantmateriet 2008). For buckwheat and strawberry, we used CORINE data from 2006 (European Environment Agency: http://www.eea.europa.eu/data-and-maps/data/corine-land-cover-2006-raster-2). For field beans we used the CORINE 2000 Land Cover Map (http://www.ceh.ac.uk/landcovermap2000.html).

Data analysis and statistics

Landscape effects on bee richness and visitation

Because different organisms act in and react to the landscape at different spatial scales, it is necessary to find a suitable scale at which to measure the surrounding landscape (Steffan-Dewenter et al., 2002; Henry et al., 2012). Before exploring any significances, we ran models for each variable with each of the different radii (0.5–3 km) at which the landscapes had been measured. Hence, for each crop we regressed percentage of agricultural area against pollinator richness and abundance at different radii, and identified the radius that explained the highest proportion of variance (highest R2). For species richness, all crops showed the highest R2 at a radius of 0.5 km, while abundance was best explained at a 1 km radius with the exception of field bean bee communities, which also responded to a larger scale (1500 m). We performed joint models for all crops at 0.5 and 1 km radius for richness and visitation abundance, respectively. Bee species richness showed a similar relation to landscape complexity for all crops, and this permitted us to include ‘crop’ as random factor in the model to investigate the general influence of landscape on richness. Visitation abundance, however, followed contrasting trajectories in relation to landscape depending on the crop. We therefore included in the model crop and its interaction with landscape as fixed effects. Pollinator abundances were centered and scaled to a mean of zero and a deviation of one within each crop. Visual inspection of rarefaction curves showed that we did not capture all species richness in most sites, therefore richness values should be interpreted as relative richness detected with an equal sampling effort. However, as real richness can be easier to detect in sites with higher pollinator abundance, we also ran the richness model for rarefied species richness at the minimum sampling size levels for each crop (Gotelli & Colwell, 2001).

Yield quantity and quality

We first correlated fruit set (or seed set for buckwheat) with yield for each crop. While we expect both to be correlated (i.e., plants with more fruits or seeds, should also have higher yield), this correlation can be stronger or weaker depending on the crop studied. Block was nested within site and included as random factor in all models. Second, we constructed one mixed effect model with yield as the response variable. In order to analyze all crops in the same model, yield and pollinator visitation abundance were centered and scaled to a mean of zero and a deviation of one within each crop. We used pollination treatment, species richness, total visitation abundance, landscape and the interactions of treatment with the other three variables as predictors. Landscape was investigated at 0.5 and 1 km radius with similar results and so only models at 1 km are shown. Block, nested within site, nested within crop was included as a random factor in all models. We checked that different crops do not present different responses by comparing AICc of this model to a model incorporating total visitation as a random slope. A significant interaction with treatment would indicate that the factor had an effect on yield only in the open treatment. To account for heteroscedasticity, we added a constant variance structure (varIdent function in package nlme, R) in which the variance was independently specified for each crop (Cleasby & Nakagawa, 2011).

We also checked if yield and quality were affected by the pollination treatment for each crop. Each crop was analyzed independently due to different quality measurement units and also because there was no homogeneous response among the crops. Block nested within site was included as a random factor in all models. In this case, we tested only for the effect of the pollination treatment, without including the interactions with species richness, visitation abundance, or landscape context due to sample size limitations. For buckwheat, we used block as a constant variance function to control for the different heteroscedasticity among blocks. The package nlme in R was used to fit all models (Pinheiro et al., 2011). Residual plots where used to check for normality and standardized residuals for heteroscedasticity.

Results

Landscape effects on bee richness and visitation

Pollinator species richness ranged from 2 to 26 species per site (Table 1). The flower visitors of all crops were highly dominated by one or two species of pollinators, in most cases managed honeybees. In field beans, the dominant species were bumblebees; Bombus terrestris/lucorum complex, followed by B. hortorum and B. lapidarius (Fig. 2). Simple landscapes had consistently lower species richness in all crops (GLMM: F1,35 = 5.39, P = 0.02; Fig. 3A). All crops responded similarly (slope ± SE = −8.43 ± 3.63), but with different intercepts (field bean = −8.39; buckwheat = 1.90; oilseed rape = 3.14; strawberry = 3.35).

Figure 2 Total number of visits recorded per pollinator guild in each crop.

All crops received the same sampling effort (i.e., four 30 min visits to 150 m transects). Note the strong dominance of honeybees in most crops.

This trend is consistent when using rarefied species richness (GLMM: F1,35 = 3.66, P = 0.06). However, the pollinator abundance trend depended on the crop (Table 2; Fig. 3B). Visitation patterns were driven by the visitation of a single species, the managed honeybee, in all crops except for field beans (Table 2). While in most regions honeybee visits were also higher in complex landscapes, in buckwheat there were higher honeybee visits in simple landscapes. For field beans, this positive relationship between number of visits recorded and landscape was even more pronounced at larger scales when we analyze the primary pollinators, the bumblebees, alone (F1,8 = 6.44, P = 0.03 at 1.5 km radius). Honeybee visits is not strongly correlated with overall non-honeybee visits (field bean pearson r = 0.19; buckwheat = 0.47; oilseed rape = 0.51; strawberry = 0.32), and we do not detect an effect of landscape on overall non-honeybee visitation (Table 2).

Table 2 Effects of land use complexity on total visitation, honeybee visitation (field beans were excluded from the honeybee model), and non honeybee visitation.

Visitation is scaled within each crop. Both models include block nested in site as random factors. Agriculture is the proportion of arable land in the surrounding landscape of each field. The slopes and standard errors (SE) of each crop are shown.

	F-value	D.f.	P-value	
Total visitation				
Crop	3.13	3	0.04	
Agriculture 1 km	0.05	1	0.81	
Agriculture*crop	3.08	3	0.04	
Residuals		32		
Slopes	Estimate	SE		
Field bean	0.52	2.19		
Buckwheat	1.78	4.86		
Oilseed rape	−4.78	4.99		
Strawberry	−2.83	5.57		
Honeybee visitation				
Crop	2.05	2	0.15	
Agriculture 1 km	2.63	1	0.12	
Agriculture*crop	3.87	2	0.03	
Residuals		32		
Slopes	Estimate	SE		
Buckwheat	1.72	1.41		
Oilseed rape	−4.59	3.56		
Strawberry	−3.38	4.21		
Non-honeybee visitation				
Crop	0.56	3	0.64	
Agriculture 1 km	0.92	1	0.35	
Agriculture*crop	3.34	3	0.03	
Residuals		32		
Slopes	Estimate	SE		
Field bean	2.50	0.72		
Buckwheat	0.06	1.59		
Oilseed rape	−0.19	1.64		
Strawberry	0.55	1.81		

Figure 3 Landscape effects on pollinators.

Relationship of (A) pollinator richness per field and (B) total number of visits per field with landscape complexity (% agriculture) at the appropriate radii. Each crop individual trend is plotted in a different color. Total visits are scaled within each crop.

Yield quantity and quality

Fruit or seed number per plant were in all cases positively correlated with yield (measured as weight per plant). However, the correlation was stronger in some crops than others (oilseed rape: R2 = 0.95, P < 0.0001; field bean: R2 = 0.90, P < 0.0001; strawberry: R2 = 0.61, P < 0.0001; buckwheat: R2 = 0.67, P < 0.0001).

Open pollination increased yield for all crops (field bean estimate = 16.42 ± 3.30 g/plant, df = 67, t = 4.97, P = 0.03; buckwheat estimate = 42.44 ± 8.27 g/plant, df = 24, t = 5.12, P < 0.001; oilseed rape estimate = 0.87 ± 0.38 g/plant, df = 69, t = 2.22, P = 0.03; strawberry estimate = 2.16 ± 0.41 g/plant, df = 67, t = 5.30, P < 0.001; Fig. 4). When analyzing all crops in combination, we did not detect an interaction between treatment and species richness, which indicates that higher richness does not increase yield in any of the treatments. However, total visitation rate increased yield in both treatments (Fig. 5A) and the response was consistent among crops as indicated by the fact that allowing the variation in the slope of each crops do not improve the model (Δ AICc between competing models = 15). Interestingly, landscape complexity measured as % of agricultural land (both at 0.5 or at 1 km) also showed a significant interaction with treatment, indicating that simpler landscapes had lower yields in the open pollinated plants. However, the trend for net-bagged plants was reversed (Table 3; Fig. 5B).

Table 3 Effects of open pollination vs pollinator exclusion treatments, visitation and landscape context on yield.

Data for four entomophilous crops grown over 10 fields in Europe (buckwheat, field bean, spring oilseed rape and strawberry). Yield and visitation are scaled within each crop. Block, nested in site, nested in crop are included as a random factor. Agriculture is the proportion of arable land in the surrounding landscape of each field. The slopes and standard errors (SE) of each treatment level are shown.

	F-value	Df	P-value	
Pollination treatment	51.51	226	<0.001	
Pollinator richness	0.37	27	0.547	
Total number of visits	6.65	27	0.015	
Agriculture 1 km radius	0.01	27	0.946	
Treatment*Pollinator richness	0.01	226	0.973	
Treatment*Total number of visits	0.15	226	0.701	
Treatment*Agriculture	9.67	226	0.002	
	Estimate	SE		
Slope visits net	0.33	0.13		
Slope visits open	0.28	0.21		
Slope agriculture net	0.65	0.54		
Slope agriculture open	−0.53	0.91		

Figure 4 Pollinator contribution to yield.

Overall yield per plant (A, C, E, G) and quality (B, D, F, H) with pollinator exclusion (Net) and open pollination (Open) for each crop. Black dots are the mean values reported in the text, and the boxplots reflects the distribution of the data. Yield is measured in seed weight per plant (g) for all crops except strawberry, which was measured as fruit weight per plant (g). Commercial grades of 1 and 2 are marketable, while grade 3 is considered non marketable.

Figure 5 Visitation and landscape effects on yield.

Interaction plots showing the relationships of (A) yield per plant and total visitation and (B) yield and landscape complexity for pollinator exclusion (open circles, dotted line) and open pollination (black circles, solid line). Total visitation and yield are scaled to a mean of zero within each crop.

In addition to quantity, the quality of oilseed rape, buckwheat and strawberry increased in the open pollination treatments (oilseed rape: oil content estimate = 1.28 ± 0.31%, df = 39, t = 4.18, P < 0.001; chlorophyll content estimate = −4.15 ± 1.76 ppm, df = 39, t = −2.37, P = 0.02; buckwheat: percentage of filled seeds estimate = 0.08 ± 0.01%, df = 12, t = 6.35, p < 0.001; strawberry: commercial grade estimate = −0.32 ± 0.06, df = 67, t = −5.36, p < 0.001). On the other hand, the nitrogen content of field beans did not increase on open-pollinated plants (estimate = −0.10 ± 0.08%, df = 37, t = −1.16, p = 0.25; Fig. 4).

Discussion

Four economically important entomophilous annual crops in Europe demonstrated highly different degrees of insect pollination dependence. When open pollinated, mean yield increases ranged from 18 to 71% depending on the crop. Three of these crops are listed as having a “modest” positive impact by animal pollination in the comprehensive review by Klein et al. (2007). However, despite being in the same category, oilseed rape and strawberry increased around 20%, while field bean reached a 40% increase in yield from average levels of insect pollination. The fourth crop, buckwheat is listed as having a large positive impact by animal pollination, in line of our reported 71% increase. The review by Klein et al. (2007) is currently the best available, most up to date source of animal pollination dependence on crops, but our data highlight a disparity of results among crops listed under the same category. Our quantitative data on animal pollination dependence provides a first step to depart from the uncertainty embedded in a categorical approach. For example, dependence on animal pollination can change by variety and region. Recent reports show variability in pollinator dependence between 0 and 30% among varieties of oilseed rape (Stanley, Gunning & Stout, 2013; Garratt et al., 2013). While we were able to standardize variety for most studied crops, strawberry fields were planted with four different varieties and the presented data should be seen as an average across those varieties (but see Klatt et al., 2014).

As expected, we found that fruit or seed number per plant was positively correlated with yield measured as weight of the marketable product per plant. However, this correlation was rather weak (r2 ∼ 0.60) for both strawberry and buckwheat. This indicates that for these crops, the total fruit or seed weight was quite variable among plants with similar fruit or seed numbers. Indeed, for strawberry, the size of the receptacle is directly related to the number of fertilized achenes, while for buckwheat the proportion of filled seeds can vary considerably and is a major component of yield besides fruit set. While previous research has focused mainly on exploring the effects of pollinators on fruit or seed set (e.g., Garibaldi et al., 2011; Garibaldi et al., 2013), which is a more direct measure of plant reproduction, yield has the potential to better reflect economic value (Bommarco, Marini & Vaissière, 2012; Klatt et al., 2014), and hence, farmers’ interest. For example, while less than 20% in mean yield increase may seem as a modest advantage from the plant perspective, for the farmers it can translate into a substantial difference in revenue.

Similarly, we report that the yield quality component is enhanced to different extents by open pollination in three out of four crops. For buckwheat, strawberry, and oilseed rape, quality is directly linked to the pollinating activity of insects. We find this despite the fact that the measure of quality and underlying mechanisms are specific for each crop, and largely unrelated among crops. Empty seeds in buckwheat accumulate little or no starch (Björkman, 1995). The shape of strawberries is directly related to a complete pollination of all ovules, resulting in a homogeneously pollinated fruit (Zebrowska, 1998). For oilseed rape, the plant allocate more oil resources to well pollinated seeds. In contrast, for field beans, the nitrogen content in the seeds was not affected by insect pollination. Other factors such as soil fertility and availability of the appropriate N-fixing bacteria (Rhizobium spp.) may play a more important role for field beans (Köpke & Nemecek, 2010). However, note that we detected no trade-off between yield and nitrogen content of the seeds, as plants with more seeds did not have lower nitrogen content. Hence, the overall protein yield (i.e., nitrogen content at the plant level) was increased with open pollination.

The treatment with netted flowers gives us estimates for the extreme cases where pollinators are completely absent, and we show that the current levels of pollination are insufficient to increase yield in the open pollinated treatment in all landscapes. As previously reported, we confirm that agricultural intensification has a drastic effect on bee species richness (Ricketts et al., 2008; Garibaldi et al., 2011). However, total visitation does not always follow the same pattern as richness. This is the case for buckwheat and field bean, where fields presenting higher total visits were located in simple landscapes. For buckwheat, most of the visits in complex landscapes were due to increased honeybee densities managed for pollination. Unfortunatelly, there is no detailed information on where hives were placed in the landscape by local beekeepers as the hives were primarily put out for honey production, rather than pollination services. In field beans we found that bumblebees responded positively to agricultural simplification, noting, however, that even the more simple field beans landscapes contain a fair amount of semi-natural habitats. Overall, we found a general positive relationship between total visitation rates and yield, but not with species richness. If the remaining species that thrive in intensively cultivated agricultural areas, including the managed honeybee, are effective pollinators, yield losses can be partly decoupled from losses of species (Bartomeus & Winfree, 2013). However, our approach does not allow us to test if current pollinator levels reach the maximum achievable yield under optimal pollination conditions.

A recent global meta-analysis highlights the role of wild species in crop systems (Garibaldi et al., 2013). The flower visitors of three out of four crops were clearly dominated by honeybees (Fig. 2) and hence, are likely to be key pollinators for those crops. Garibaldi et al. (2013) show that an increase in wild insect visitation enhanced fruit set by twice as much as an equivalent increase in honeybee visitation. While this is generally the case in our target crops (three of which were included as part of Garibaldi’s synthesis), the numerical advantage of honeybees in European agricultural landscapes needs to be acknowledged when calculating their total contribution to pollinated plants (e.g., as done in Winfree et al., 2007; Rader et al., 2009). However, increasing or maintaining high pollinator diversity can enhance yield quantity and stability by improving the pollination efficiency of honeybees (Greenleaf & Kremen, 2006) and reduce the risk of pollination failure due to climate change (Rader et al., 2013; Bartomeus et al., 2013b), or environmental disturbances such as extreme weather events (Brittain, Kremen & Klein, 2012).

Overall, we also found a weak negative effect of land use intensity on yield (Garibaldi et al., 2011, but see Ricketts et al., 2008), but this was not directly mediated by increased pollinator visitation by itself, because the correlation between pollinator total visits and the proportion of agricultural land in the landscape was weak. The yield of experimental plots with net bagged flowers also increased in sites with more pollinators (Fig. 5A). This suggests that other environmental or biotic factors correlated with insect visitation may have been operating simultaneously. The release of airborne pollen by foraging bees could be such a factor (Pierre et al., 2010).

In order to make efficient management decisions and increase our power to predict the actual benefit from pollinators in a certain farming situation, we need to estimate the combined contribution of multiple ecosystem services and agricultural inputs (Boreux et al., 2013), as they may be influenced differently by landscape characteristics or have non-additive interactions among them (e.g., Lundin et al., 2013; Martin et al., 2013).

Information on the benefit delivered by pollinators to yield quantity and quality in relation to landscape context provides an important baseline for this work.

Supplemental Information

Supplemental Information 1 Data used for the analysis

Insect richness, visits per guild, mean yield and quality are reported for each site. See text for details. Data is in csv format, skip 2 first lines when importing in a data analysis program.

Click here for additional data file.

We thank H Dathe, G Else, R Fonfria, S Iserbyt, M Kuhlmann, G Le Goff, D Michez, H Mouret, A Müller, S Patiny, A Pauly, P Rasmont, S Risch, M Schwarz, R Theunert, C Waldemar, and P Williams for bee identifications to species, V Zaldo for GIS assistance and V Gagic for statistical discussions.

Additional Information and Declarations

Competing Interests

Author Contributions

Data Deposition

The authors declare there are no competing interests.

Ignasi Bartomeus analyzed the data, wrote the paper, prepared figures and/or tables, reviewed drafts of the paper.

Simon G. Potts, Ingolf Steffan-Dewenter and Michal Woyciechowski conceived and designed the experiments.

Bernard E. Vaissière conceived and designed the experiments, wrote the paper.

Kristin M. Krewenka, Thomas Tscheulin, Stuart P.M. Roberts, Hajnalka Szentgyörgyi and Catrin Westphal performed the experiments.

Riccardo Bommarco conceived and designed the experiments, performed the experiments, reviewed drafts of the paper.

The following information was supplied regarding the deposition of related data:

Insect richness, visits per guild, mean yield and quality are reported for each site in Supplemental Information 1.

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
