# Peer review of "Contribution of insect pollinators to crop yield and quality varies with agricultural intensification"

_PeerJ, doi:10.7717/peerj.328_

## Round 0.1 · original submission · Minor Revisions

Dear Ignasi,
your ms has now been reviewed by two referees who agree it is a relevant paper worth of publication. However, they have some concerns that I would like you to consider when resubmitting your work to PeerJ. Basically, I would like you consider the following:
1. Clarify the set up of the pollination experiments. I agree with one of the referees that a figure would make clear the treatments and number of replicates.
2. Be clear with the distinction betweeen fruit set and yield as these are two different things. Both referees have pointed out this concern and I agree with them.
3.Describe in more detail how honeybees were managed (number of hives...) as this will allow understanding beteer the experimental set up.
4.Analyze how honeybees and native bees interact and show results on visitation rates separate for them.
5. In the statistical analysis, you should justify why you include the predictor variables in two models and also why you consider crop species as a random factor in models 1 and 2.
6. Rarefaction curves to assess the completeness of your sampling procedure would be desirable.
7. The sentence on pollen limitation needs also clarification. I agree with the referee that to assess maximum pollination you would have needed to perform hand-pollination additions.

I hope you find all comments useful and look forward receiving a revised version of your ms.

Reviewer 1 ·

Basic reporting

This article meets all the PeerJ criteria. The introduction and background is appropriate and comprehensive, and firmly situates the paper within the ecosystem services literature. Although the authors do not specifically delineate hypothesis, it is clear they expect crop visitation to decline along a landscape gradient. It may enhance clarity to specifically outline hypotheses.

Experimental design

The study has a robust experimental design, with 10 sites in each of 4 crops along a landscape gradient. Sampling was replicated within each field. The description of the block design for the pollination experiments is somewhat unclear and I would suggest a figure in the supplement or paper to depict it and improve replicability of experiment.

Validity of the findings

Findings meet journal standards. The study showed that pollination varied by crop and landscape context, though honeybees were consistently dominant in almost all landscape and crop types. This builds on past studies and meta-analyses, while providing a deeper understanding of the variability of pollination, while also highlighting the yield and quality factors important to growers.

Additional comments

This is a relevant and interesting paper. The main concern I have is that you did not describe how honeybees were managed, therefore it is challenging to interpret how their visitation rates were affected by the different landscape contexts you studied. It would add clarity if you addressed this. Additionally, honeybees may interact with native bees, it would be interesting to examine this in your analyses. Also, you look at total visitation and honeybee visitation- what about just wild pollinator visitation?

Specific comments:

5|75 Do you mean “homogenous” instead of “monotonous”

5|82 add commas . . diversity of response can, in some cropping systems, buffer . . .

5|84 I am not sure what the term “spatially disassociated” means

7|128 It is hard to visualize the design as described by this statement “Each block had two treatments with one plot per treatment”. Perhaps a figure in paper or supplement of the block design would assist in replication of this experiment and clarification of your methods.

9|159 That is more than half of your buckwheat sites. Were they evenly distributed along the landscape gradient? Do you think these results accurately reflect the trends in buckwheat pollination? You might want to bring this back up in the discussion if the sites were in a particular landscape type.

10|202 Not clear how fruit set and yield are different, thus also unclear why you need to find out whether they are correlated. Can you describe how you measured them in more detail? (It would also help to clarify the paragraph in the discussion that focuses on this issue)

11|206 Could you include honeybee and native bee visitation as separate variables in your model, and also look for an interaction between the two? I know you did the analyses separately, but it could be interesting to look for a signal that higher native bee abundance increased honeybee activity (as Greenleaf and Kremen did in their 2006 PNAS paper).


12|233 What does “visitation patterns were driven by honeybees” mean? Were most of the visitors honeybees, or did honeybees affect the way other bees in the system visited the focal crop? Additionally, here, or in the discussion, it might be helpful to describe the number of hives typically brought in for each crop. Did it differ in simple or complex landscapes, or was it always the same? If it was the same, then the honeybees might be foraging on different crops . . . Some kind of explanation beyond landscape context but related to honeybee management seems necessary. Or possibly describe in the study sites section.

15|303 Why is it interesting to note that? This sentence seems somewhat disconnected from previous paragraph

16|336-8 Again, this could be due to the number of managed honeybee hives.

Reviewer 2 ·

Basic reporting

In the manuscript entitled “Contribution of insect pollinators to crop yield and quality
varies with agricultural intensification” the authors quantify the degree of dependence of insects for four different crops in Europe. They analyze the contribution of insect’s pollination to yield quantity and quantity and relate the yield with richness and abundance of insect visitor and with landscape context (the proportion of arable land within each crop). The article is clearly written and includes sufficient information in the introduction that demonstrates that the topic is under current study, with relevant and recent literature cited. Figures are relevant (I added comments to improve the clarity of the results on some of them).

Experimental design

The experimental design is clearly stated. I have some minor comments that hope to improve the clarity of the manuscript.
Line 104: Please provide the scientific name of the cultivated species for oilseed rape and for strawberry.
Table 1: Please provide the range of complexity for the two main radii analyzed for each crop.
Please clarify in the methods and statistical analysis the difference between fruit set and yield and the units. In most of the studies of pollination fruit set is the number of fruits/flowers, thus is a rate. I was expecting that the relation between fruit set and yield was fruits/flower vs fruits/ha (or seeds/ha). Please clarify this in the methods and results. It is not obvious.
Do the authors have any measure (e.g. rarefaction curves) to know is sampling procedure was enough to detect properly species richness within each site?
Why the authors did two different models with the same response variable instead of only one model that included as predictors all the variables (spp richness, abundance, pollination treatment, landscape complexity and interactions)?

Validity of the findings

The conclusions are appropriately stated, connected to the original question investigated. Some results could be more informative, thus I added some comments:
In the results on landscape effect on pollinators it would be useful to have the estimated slopes for each crop as well as the general slope when appropriate (please also provide confidence intervals for the estimators). In figure 2 I think that in order to better compare within crops standardized graph could be useful.

I am not sure about the appropriateness of considering the crop species as a random factor for models 1 and 2. The authors want to do inferences with each crop, as shown in Figure 3 and as stated in the conclusions. If the authors leave as they did the model 1 with crop as a random factor, then they technically cannot do the figure 3 (A, C, E, G). In figure 4 instead they only show the general results that are consistent with the model. But I would like to see how different is the open and exclusion treatment with the landscape complexity for each crop. I can be informative to know the difference (the estimated difference in the model) for the pollination treatment for each crop; that is how much insect pollination increase yield. I think that this information can be in the text (it is shown in figure, but the authors state in the introduction how important is to have this information for different crops)

Figure 3 please provide the units for yield.
Line 251-257: please check the symbols. Why for some crops the estimated difference between open and exclusion is positive (that is there is an increase with pollinators (e.g. oil content in oilseed rape) while for strawberry is negative?
In the discussion: the authors state “The treatment with netted flowers gives us estimates for the extreme cases where pollinators are completely absent, and we show that the current levels of pollination do not suffice to maximize yield in the open pollinated treatment in all landscapes”. For me it is not clear this. In order to have the estimations of a maximum yield authors should have done another control that is the maximum levels of pollination (with the addition of pollen). Open pollination can be in the middle between the maximum possible and the levels of reproduction without insects (the netted flowers).

The authors could also state others likely reasons for the increased yield in the netted flowers. There could be that the manipulation of bags increased the levels of self -pollination?

Additional comments

Other minor comments:
The sentence in the line 39-40 needs a reference.
Line 310: correct Ricketts.
Be consistent with the use of complexity of the landscape and proportion of arable land (in tables in the results authors use agriculture, while in the text they use complexity).

---

## Round 0.2 · accepted · Accept

Dear Ignasi,
Thank you for revising the ms, which I think it has notably improved after considering the comments suggested by the two referees.